# Analytical Investigation of Iron-Based Stains on Carbonate Stones: Rust Formation, Diffusion Mechanisms, and Speciation

**DOI:** 10.3390/molecules28041582

**Published:** 2023-02-07

**Authors:** Rita Reale, Giovanni Battista Andreozzi, Maria Pia Sammartino, Anna Maria Salvi

**Affiliations:** 1Chemistry Department, University of Rome ‘La Sapienza’ Piazzale Aldo Moro 5, 00185 Rome, Italy; 2Earth Sciences Department, University of Rome ‘La Sapienza’ Piazzale Aldo Moro 5, 00185 Rome, Italy; 3CNR-IGAG c/o Department of Earth Sciences, Sapienza University of Rome, 00185 Rome, Italy; 4Science Department, University of Basilicata, Viale dell’Ateneo Lucano 10, 85100 Potenza, Italy

**Keywords:** iron stains, mechanism of rust formation, iron speciation, Mössbauer spectroscopy, XPS, SEM/EDS, OM

## Abstract

In cultural heritage, unaesthetic stains on carbonate stones due to their close contacts with metals are of concern for the preservation of sculptures, monumental facades and archeological finds of various origin and antiquities. Rust stains made up of various oxidized iron compounds are the most frequent forms of alteration. The presence of ferric iron on rust-stained marble surfaces was confirmed in previous studies and oriented the choice of the best cleaning method (based on complexing agents specific for ferric ions). However, the composition of rust stains may vary along their extension. As the corrosion of the metallic iron proceeds, if the oxygen levels in the surroundings are low and there are no conditions to favor the oxidation, ferrous ions can also diffuse within the carbonate structure and form a variety of intermediate compounds. In this study, the iron stains on archeological marbles were compared with those artificially produced on Carrara marbles and Travertine samples. The use of integrated techniques (optical and scanning electron microscopy as well as Mössbauer and XPS spectroscopy) with complementary analytical depths, has provided the overall information. Rust formation and diffusion mechanisms in carbonates were revealed together with the evolution of iron speciation and identification of phases such as ferrihydrite, goethite, maghemite, nanomagnetite, and hematite.

## 1. Introduction

The use of carbonate stones in cultural heritage has always interested both architecture and statuary. Unfortunately, stone artifacts (especially those exposed in outdoor environments), despite their resistance, show various forms of degradation, due to chemical, physical or biological interactions recurring over time [1,2]. Among these, the colored stains formed on stone surfaces when in contact with metals or alloy, such as iron-based stains on carbonate stones or bronze statues on stone pedestals [3], are of considerable impact, not only aesthetic. In particular, iron-based stains imply the diffusion of corrosion products and related risk of cracking [4] and can originate from two distinct sources. The first source is the oxidation of the iron coming in contact with the stone from the water supply, mounting nails or screws, decorative elements, gratings, etc. A secondary source is the oxidation of the iron minerals present in the stone, such as pyrite or marcasite or other iron compounds. The first source of stain is the most common, as metallic iron, when exposed to the natural environment (e.g., acid rain, humidity, and temperature variations), is subject to corrosion and produces a colloidal phase of ferrous or ferric hydroxides, which propagate both on the stone surface and inside, through the boundary grains. Subsequently, the dehydration of these oxyhydroxides leads to the formation of different hydrated oxides, which are less soluble and more difficult to remove [5]. Indeed, the choice of the correct cleaning treatment is crucial for the need to combine the best stain removal, based on the solubility of the specific iron phases, with the restoration of the historical material “without any structural damage”, in compliance with the theory of cultural heritage conservation [6]. Consequently, to realize an effective cleaning procedure, a better knowledge of the stain compositions is required.

Although the pure phases of iron oxides are well known, as are their topotactic transformations [7], the rust composition varies with the different environments that have determined its growth. Given that rust is mainly found to be composed of nanocrystalline ferrihydrite (Fe^3+^_10_O_14_(OH)_2_), lepidocrocite (γ-FeOOH), goethite (α-FeOOH), and/or hematite (α-Fe_2_O_3_) [8,9], the formation mechanisms of these phases within carbonate stones are largely unknown and their interactions with carbonate matrix may be of critical relevance. In fact, iron stains not only disturb the aspect of stones but can also cause chemical-physical damage, as the oxidation of metallic iron to give oxidic phases leads to an increase in volume of the layer surface, due to oxygen penetration in the crystalline lattice [10]. In the case of iron oxidation inside the stone artifacts (bars, rods, or pins), the released corrosion product is higher in volume than the original metal [11], and the ratio between the volume of expansive corrosion product and the volume of iron consumed in the corrosion process is called the “rust expansion coefficient” [12]. The resulting increase in volume associated with the formation of corrosion products gradually induces tensile stresses within the surrounding carbonate matrix and may strongly damage the stone artifacts.

In previous work [13,14,15] using surface techniques, we easily identified iron (III) compounds on the surface of stained marbles and carbonate stones. We thus proposed as suitable chelators for their removal the natural compounds glutathione and deferiprone dispersed into alginate gel and two proteins of the transferrin family (lactotransferrin, Ltf and ovotransferrin, Ovt) supported by cellulose pulp. The results, very satisfyingly, showed the efficacy of the selected chelating agents, all known for their affinity for ferric ions in vivo, antimicrobial activity and selectivity, ensuring safety for the carbonate structure and for the operators. Each chelant has proved the best cleaning action when dispersed in the most suitable support to be spread on the surfaces to be treated and removed after the given contact time. Overall, the proposed green methodologies are suitable to remove iron stains from carbonate stones as testified by visual inspection, optical microscopy, and XPS, comparing the Fe/Ca ratio before and after cleaning. Also, it was seen that iron (III) removal also always implies the removal of surface contaminants associated with rust deposits, eventually bioactive [16]. The cleaning efficacy of the removal is strongly dependent on (a) chelant properties and bioactivity, (b) volume of water retained by the support, which determines the number of iron complexes to be therein dissolved, and (c) type and entity of the rust deposit and its lateral and in-depth extension. In practice, subsequent applications were foreseen to remove the rust patches completely, leaving the layers underneath to be exposed outside following the first application. Clearly, within the context of cultural heritage, particularly for archeological artifacts, different iron oxidation states can be encountered at the subsurface and boundary interfaces; therefore, highly differentiated treatments may be required.

In this work, we investigated iron stains of (a) historical archeological white marble and (b) artificial iron-stained surfaces of Carrara marble and Travertine, suitably sized for laboratory tests, using complementary analytical techniques, such as optical microscopy (OM), scanning electron microscopy (SEM-EDS), X-ray photoelectron spectroscopy (XPS), and Mössbauer spectroscopy. To better understand the development of rust on the carbonate matrices from these comparisons, the laboratory approach (see Section 4) was simply based on the seasonal monitoring of the corrosion products produced on Carrara marbles and Travertine samples by metallic iron “placed in contact”, so as to recall the degradation phenomena affecting ancient marbles over time under the recurring action of atmospheric agents in outdoor and indoor conditions.

The main outcome was the definition of the different states of iron oxidation in rust stains, the identification of the corresponding corrosion products, and insight on their evolution over time.

## 2. Results

### 2.1. Optical Microscopy (OM) and Electron Scanning Microscopy Coupled with X-ray Microanalysis (SEM/EDS)

Optical microscopy allowed the choice of the most significant historical iron-stained samples to be used for further analytical investigations. These samples (NS1, 19th century A.D., and NS3 and NS8, 4th century A.D., Figure 1a, b, and c, respectively) show the characteristic pigmentation of iron compounds, due to the oxidation of pins or brackets, penetrated within the carbonate substrates visible in cross sections in reflected light (Figure 1d–f, respectively). Fractures, as in Figure 1a, can be attributed to the insertion of the pin and/or to the volume increase associated with metal oxidation. In all samples, the area surrounding the iron element shows a compact and brown coloration that penetrates the grain boundaries inside the carbonate matrix, drawing reddish brown and yellow–orange bands (Figure 1d–f) similar to cyclic precipitations of iron oxides, known as Liesegang rings [17,18].

OM (with analysis in polarized light) showed important differences in the petrography of the three historical marble samples (Figure 2).

NS1, a white calcitic marble, shows an isotropic microstructure, with lobed intergranular sutured contacts and widespread triple junctions, with grain size between 0.55 to 1.03 mm (Figure 2a). Xenoblasts of calcite are visible, with evident rhombohedral and poor lamellar polysynthetic geminations, rare and very small opaque minerals.

NS3, a calcitic marble, shows an isotropic microfabric with irregular shapes and grain size between 0.2 and 0.6 mm, with sutured/lobated grain boundaries (Figure 2b).

NS8 shows a microfabric with size of 0.15–0.35 mm, with intergranular contacts, mainly interlobates, although occasionally pseudolinear, and with an abundance of triple junctions (Figure 2c). In the same sample, in addition to the calcite, few opaque mineral phases are visible. The texture is mainly isotropic; however, there are parallel bands with a thickness of about 1 mm, while the average size of the crystals is smaller (0.06 and 0.15 mm).

In all these samples, the precipitation of iron oxides takes place mainly along the fracture surface, the cleavage planes, and at the grain boundary cavitations, with a higher concentration where the grains are very small (Figure 2). As expected, when the distance from the metal element increases, the concentration of iron oxide hydroxides progressively decreases.

The artificially stained samples (see staining procedure in Section 4), unlike the historical samples, show in the cross section (observed in reflected light) a superficial growth of iron oxides a few micrometers in thickness in both the Carrara marble and Travertine. The diffusion within the calcite matrix reaches about 1 mm maximum in Carrara marble, with a decreasing color intensity, while for Travertine, diffusion filled mainly the voids (Figure 3).

More information about the iron compounds and their distribution was obtained by a detailed observation of thin cross sections by SEM/EDS in a portion of the sample where the carbonate matrix is less compact, more porous, and carious, with uneven distribution of iron compounds. The line scan on the NS1 sample shows the distribution profile of Ca and Fe (Figure 4): on the opposite sides, there are the highest concentrations of Fe (left, corresponding to the external part) and Ca (right). The concentration of Fe falls below half at about 250 μm, Fe remains the most abundant elements up to about 800 μm, then there is an area in which Ca and Fe alternately reach similar concentration, and finally Ca becomes prevalent at a depth of about 1.7 mm.

The observation of the thin NS3 section shows how the calcite was substituted by iron-rich phases (see lighter zones in Figure 5a). The elemental analysis of point 1 (Table 1) reveals low iron concentration (0.3 wt%). The carbonate matrix is still dominant, but there is a noticeable penetration of the iron along the cleavage planes [19]. In point 2, a significant precipitation of Fe (Fe 55.6 wt%, Ca 4.3 wt%) increases the intergranular pore spaces surrounding the residual carbonate grains.

In Figure 5a, the red arrows highlight areas with a very porous appearance that represent the contact between the “healthy” portion of carbonate matrix and the diffusion fronts of iron, which irregularly penetrates the calcite body along the intergranular contacts and replaces it. Small fragments of calcite, residual, or precipitate from dissolution process are visible. In Figure 5b, many voids appear partially filled with different iron phases [20] that may be morphologically identified as microporous colloform goethite (dark gray) and lamellar hematite crystals (light gray) [21]. Confirmation of all the natural samples by XRD was impossible because of the small quantity of iron phases in the powder samples, which were below the diffractometer detection limit (about 1 percent of a mineral’s content).

### 2.2. XPS Analysis

A systematic characterization of the artificial stains produced on marble samples was undertaken by collecting the samples gently scraped as powders and homogenized for XPS analysis, with the aim of determining the chemical states of iron and other elements composing the rust deposited as a function of marble type, surface and inner corrosion, and environmental conditions. A well-established curve-fitting procedure was used [22,23].

XPS spectra were processed and resolved in their peak components, thus allowing chemical state assignments (using corrected positions in binding energy, BE eV) and relative percentage composition (by the normalized peak area ratios). From the large number of experiments on Carrara marbles and Travertine (whose comparative characterization is still in progress), in this work we report the curve-fitted Fe 2p region for the stained Carrara samples (Figure 6) with long exposure outdoors (spring/summer) and all the detailed regions acquired (Table 2).

The curve-fitting results with the proposed identification of the corrosion products formed on the external surface and at the carbonate interface of the iron-stained Carrara marbles were considered the most suitable to be compared with those found in archeological samples by other techniques, in particular Mössbauer spectroscopy (vide infra).

Both Table 2 and Figure 6a indicate the presence of only Fe^3+^ ions in the external growth layer, in agreement with literature data and XPS results previously reported [15], while Fe^2+^ compounds and other Fe-containing phases in reduced chemical states (more difficult to identify) are co-present only in the interface zone, Figure 6b (vide infra).

In more detail, for the external layer, the Fe2p curve-fitting parameters matched exactly those of the standard α-Fe_2_O_3_ acquired in the XPS laboratory [15], represented by four peaks for each 2p3/2 and 2p1/2 doublet component, comprising one main peak, two split multiplets (MS), and one shake-up (SU) satellite (Figure 6a). In contrast, for the sample collected at the interface, in addition to the Fe^3+^ doublet still represented by the multiplet and shake up satellites, two extra doublets, with their main 2p3/2 components at significantly lower BE (708.2 eV and 704.7 eV), were required to properly fit the whole Fe2p envelope (Figure 6b). To also reduce computational complexity, both components were tentatively represented by only one doublet (2p3/2 and 2p1/2 separated by = 13 eV) with a large FWHM (full width, half maximum) to match the experimental line, without the need to split them into multiplet and shake-up satellites.

Considering in [24] the superimposed spectra of Fe^3+^, Fe^2+^ well-defined oxides and Fe metallic, also reported in [15], and their relative BE positions well in agreement with literature data [25,26,27] and the XPS online database [28], the two extra peaks of lowest BE found at the interface are not straightforwardly identifiable, and further consideration comprising all the other detailed regions, elaborated by curve-fitting (Table 2), is required.

In support, we recall the XPS studies of stratified systems, intercalated with organic and inorganic compounds, of worn surfaces, oxide nanoparticles, coupled inorganic nanoparticles–nanocarbon for ultrafast batteries, encapsulated iron carbide catalysts, etc., just to mention some of the relevant publications [29,30,31,32,33,34,35].

Experimental results highlight the local information provided by XPS for the element under investigation, dependent on chemical bonds, aggregation sizes, and surrounding environments. Of relevant importance are the chemical shifts reported for intercalated Fe^3+^ and Fe^2+^ ions found at 709.5 and 707.8 eV, respectively, confined Fe^2+^ carbides at 708.2 eV, and a variety of signals ranging from Fe^3+^ to metallic Fe, with the lowest BE feature of Fe 2p3/2 due to strong metallic aggregates at 705 eV, using an internal standard similar to ours based on aliphatic/aromatic C-C set at 284.8 eV.

Based on a complete elaboration of the considerable data set collected during a triennial doctoral research project [36], in light of the above insights, important information can be gleaned from the data reported in Table 2 for each elemental peak, chemical state(s), and relative intensity (peak area). The main outcome, based on peak area mass balance, taking into account the stoichiometric coefficients for the given chemical states, is as follows.

Iron oxide spread on surfaces in the form of more or less hydrated Fe_2_O_3_ does not affect the calcite structure, leaving the correct CaCO_3_ stoichiometry.

In agreement with previous work [13,14,15], iron stains are often associated with organic contaminants that enter the rust composition. Of the contaminants deposited on surfaces, the aliphatic carbons can be recognized by curve-fitting the C1s region and taken as the internal standard, set at 285.0 eV, to correct all the binding energies.

At the interface depths, iron chemical states could be the result of its intercalation within the carbonate structure and confined association. The peaks at the lowest BE side of Figure 6b can thus be assigned to a variety of iron compounds ranging from metallic and organometallic aggregation, through iron carbide and Fe^2+^ oxides prone to further oxidation once exposed to air [28,29,30,31,32,33,34,35].

Concomitantly, comparing the surface and interface Ca:CO_3_ stoichiometry, a fairly consistent imbalance was observed at the interface with excess calcium, represented as CaO in Table 2, significantly shifted 0.6 eV to higher BE than surface CaO.

### 2.3. Mössbauer Analysis

Mössbauer spectra were collected by measuring at room temperature samples made of powder scraped from the inner and outer rust layers of both archeological and artificially stained marble samples.

The samples with long outdoor exposure for one, two, three, and four seasons show very similar Mössbauer spectra dominated by the presence of a strong paramagnetic doublet, with an isomer shift of about 0.36 mm s^−1^ and a quadrupole splitting of about 0.60 mm s^−1^ (Figure 7).

According to the literature [37], this may be interpreted as due to an irresolvable mixture of the Fe^3+^ oxide hydroxides lepidocrocite, ferrihydrite, nanogoethite, and nanohematite, with the last two exhibiting superparamagnetic behavior because of an average size <10 nm (Table 3). A doublet corresponding to Fe^2+^ was tentatively added to the spectrum of the samples exposed outdoors for one season to verify if any signature of the possible presence of reduced Fe was detectable, as indicated by XPS analysis at the interface zone together with even more reduced Fe as minority components. Unfortunately, the resolution of the spectrum was so low that the presence of Fe^2+^ could be neither confirmed nor excluded (Figure 7a). A weak additional sextet, with IS = 0.33 mm s^−1^ and HF 49 T, was clearly observed only in the spectrum of the samples exposed for four seasons (Figure 7d). A likely attribution of the sextet is to ferrimagnetic maghemite or, less probably, to ferromagnetic nanohematite with average particle size >10 nm, both phases pointing to an evolution of the rusty deposits with time.

By merging the spectra obtained from the different exposures (Figure 8a), it appears evident that they are all dominated by the central quadrupole doublet, likely due to the presence of Fe^3+^ oxyhydroxides, either amorphous or low crystalline (with average size less than 10 nm). By increasing the outdoor exposure, such poorly crystalline, nanometer-sized phases of Fe^3+^ oxyhydroxides reasonably underwent aggregation processes towards relatively larger crystalline Fe^3+^ oxides, such as maghemite and/or hematite. A similar multiple signature was also observed for the sample exposed indoors, AS 25, stained for three months in a highly humid environment (Figure 8b). In this case also, the absorption bands were fitted with a central doublet (IS = 0.35 mm s^−1^, QS = 0.60 mm s^−1^), interpreted as lepidocrocite/ferrihydrite/nanogoethite/nanohematite, and two sextets (IS = 0.30 mm s^−1^, HF 48 T, and IS = 0.34 mm s^−1^, HF 50 T), interpreted as maghemite/hematite (Table 3). Other contributions due to Fe oxide/oxyhydroxide nanoparticles (emerging as relaxed sextets) are possible, but they could not be fitted with confidence at room temperature. Results obtained from sample AS25 demonstrate that Fe aggregation processes and stain formation were more rapid for the sample exposed indoors than for those exposed outdoors, even for a year. Similar results showing the stronger dependence of stain formation on humidity with respect to time were obtained in the literature [38].

Absorption spectra of the historical samples NS1and NS8 are quite similar and show both quadrupole doublets and magnetic sextets (Figure 9). In the NS1 spectrum, the two quadrupole doublets (IS = 0.31 mm s^−1^, QS = 0.75 mm s^−1^, and IS = 0.76 mm s^−1^, QS = 1.18 mm s^−1^) were considered as due to the possible presence of Fe^3+^ from low crystalline ferrihydrite and iron at a mixed valence Fe^2.5+^ from intervalence charge transfer (IVCT), respectively. Concerning the three sextets, the first, IS = 0.37 mm s^−1^, HF = 36 T, was attributed to goethite, while the other two, being very relaxed with IS = 0.41 mm s^−1^, HF = 31 T and IS = 0.59 mm s^−1^, HF = 20 T, were tentatively attributed to superparamagnetic nanomagnetite (Figure 9a and Table 3). In NS8, the two quadrupole doublets (IS = 0.35 mm s^−1^, QS = 0.67 mm s^−1^, and IS = 0.67 mm s^−1^, QS = 1.44 mm s^−1^) were attributed to the same Fe species identified in NS1, while two sextets were considered as due to goethite (IS = 0.35 mm s^−1^, HF = 36 T) and maghemite (IS = 0.35 mm s^−1^, HF = 50 T), and the third, being very relaxed with IS = 0.35 mm s^−1^ and HF = 28 T, was tentatively attributed to superparamagnetic nanohematite (Figure 9b). Rearranging the obtained pieces of information, in NS1 and NS8, iron stains are tentatively attributed to the combination of Fe^2+^–Fe^3+^-bearing amorphous/crystalline phases, such as magnetite nanoparticles, and their oxidation products, likely Fe^3+^ oxide hydroxides of higher crystallinity and larger size. Notably, the oxidation of NS8 is more advanced than that of NS1, as NS8 contains maghemite and nanohematite in place of nanomagnetite. In the literature, a similarly broad sextet pattern was described by a distribution of hyperfine fields attributed to medium-sized nanoparticles [39].

The absorption spectrum of the historical sample NS3 sample only appears to be different from the previous ones (Figure 10), though it is made of two quadrupole doublets and three magnetic sextets like them (Table 3). The two quadrupole doublets, one with IS of 0.37 and QS of 0.55 mm s^−1^, the other with IS of 0.67 and QS of 1.15 mm s^−1^, are attributed to lepidocrocite and IVCT Fe^2.5+^, respectively. One of the three sextets is well defined, with IS = 0.27 mm s^−1^ and HF = 49 T, and is attributed to maghemite (less probably to relatively large nanohematite), while the other two, being very relaxed with IS = 0.35 mm s^−1^, HF = 33 T, and IS = 0.67 mm s^−1^, HF = 20 T, are tentatively attributed to superparamagnetic nanomagnetite (Table 3).

## 3. Discussion

By comparing studies on real samples differently aged with those on artificial specimens stained at different exposure periods, we were able to make a contribution to the definition of the mechanisms of iron stain formation and its speciation, which will be useful to individuate procedures for better cleaning.

In agreement with the results of Beltran et al. (2016) [40], transport of the iron on carbonate stones mainly occurs as a colloidal solution of oxyhydroxides, which transform with time in more stable forms. In the present investigation, the transformation over time is documented by Mössbauer spectroscopy that shows both in historical and artificial stains the presence of nanometric clusters of Fe that, in the course of time, tend towards larger oxyhydroxides/oxides. Iron diffusion was shown by microscopy techniques to be dependent on structural characteristics and stone porosity. Using archeological samples, it was demonstrated that iron replaces dissolved calcite, moving through the grain boundaries along irregular paths by cycling precipitation of iron oxides. Moreover, XPS allowed discriminating the chemical composition of the iron phases diffused inside the substrate and those grown on the surface. As an example, the two samples in Figure 6 present only carbonaceous contaminants and no others. As reported in Table 2, only magnesium oxide is present, likely as an intrinsic component of the Carrara marble. The results already explained in the XPS section can be compatible with the action of water and oxygen triggering iron corrosion till a potential equilibrium is reached. Corrosion also depends on the adsorption of organic contaminants, the lowering pH of rainwater due at least to the presence of dissolved carbon dioxide, and eventually more acidic pollutants (sulfur and nitrogen oxides) on dissolution and reprecipitation of calcium carbonate and other related factors [41,42].

As expected, the actions of natural weathering, specific environmental pollutants, alternating wetting and drying cycles, and seasonal fluctuations in climatic conditions affect the stones’ integrity and the iron corrosion potential, inducing changes in the rust composition and internal diffusion. It is therefore difficult to singly identify the factors responsible for deterioration, as they can act in synergy and/or in temporal sequence with different modalities, eventually including biodegradation processes [43].

We believe, however, that research combining parallel laboratory experiments and advanced analytical techniques could progressively contribute to a more accurate characterization of each case study, the overall outcome providing the right directions for effective maintenance and conservation intervention.

## 4. Materials and Methods

### 4.1. Sampling

Several iron-stained samples (Table 4a) of white marble were collected from the archeological remains of ancient Roman baths discovered beneath Palazzo Valentini (Rome, Italy) and the warehouses of ancient marble sculptures of the Vatican Museum, according to UNI 11182 (Description of alteration forms—Terms and definitions) and UNI EN 16085 (Methodology for sampling from materials of cultural property. General rules).

The artificial stains were produced through the oxidation of iron cubes (10 × 10 mm) and nails put in contact with the surface of Travertine (Tivoli quarries) and Carrara marble samples (50 × 50 × 20 mm), both indoors in a highly humid environment and outdoors exposed on a terrace (Table 4b). The outdoor exposition was continued for one year to achieve stains from iron oxides related to one to four seasons.

### 4.2. Analytical Techniques

#### 4.2.1. Optical Microscopy

The preparation of thin and cross sections was performed according to the specifications of the UNI 10922: 2001 Standard (Ref. Recommendations CNR—ICR NorMal 10/82, Natural and artificial stone materials—Preparation of thin sections and cross sections of stone materials). The petrographic investigation was performed by means of optical polarized light microscopy (transmitted and reflected) with a Zeiss Axiolab, equipped with a Nikon D800 camera for digital image capture.

#### 4.2.2. Scanning Electron Microscopy with EDS (Energy-Dispersive Spectroscopy)

Scanning electron microscopy was performed on the samples selected from optical microscopy (NS1, NS3, NS8) using a Jeol JSM 5400 to determine the chemical composition of mineral phases through EDS of backscattering (BSE) images. To produce an electrically conductive surface for SEM, both thin and cross sections were coated using thin-film evaporation of graphite in a vacuum coater, with a thin layer of about 20 nm thickness.

#### 4.2.3. X-ray Photoelectron Spectroscopy (XPS)

XPS spectra were acquired with a SPECS Phoibos 100-MCD5 spectrometer operating at 100 W in FAT (fixed analyzer transmission) and medium area modality (spot diameter ≈2 mm), using Al Kα (1486.6 eV) and Mg Kα (1253.6 eV) achromatic radiation. The pressure in the analysis chamber was higher than 10^−^^9^ mbar (UHV, ultrahigh vacuum). The use of the double anode (Al/Mg Kα) helps to distinguish XPS signals, varying in kinetic energy with the source employed, from the X-ray-induced Auger signals, dependent on atomic relaxation following photoemission [26]. High-resolution detailed regions (channel width of 0.1 eV) were elaborated by curve-fitting using the home made Googly software [22,23] qualitatively by referring the binding energies to the acquired spectrum α-Fe_2_O_3_ (standard Aldrich) for correcting the surface charging and to literature data and quantitatively by normalizing the peaks area with appropriate sensitivity factors [26]. For XPS acquisitions, the surface sampling was performed by carefully scraping the surface of the blank and stained marbles, respectively not in contact and in contact with the iron (nails, cubes). The interface sampling was performed by immediately scraping the inner part of the marble specimens left exposed after having the outer layers removed for the previous Surface analysis. The powders collected were all stored in special inert and sealed plastic containers to avoid external contamination.

Hence, by referring to the acquired spectrum α-Fe_2_O_3_ (Sigma Aldrich Hematite 99%, St. Louis, MO, USA) and to literature data mentioned in the relevant paragraphs, the oxidation products derived from curve-fitting for all analyzed samples could be compared within the limits of XPS accuracy (±10%) [26].

#### 4.2.4. Mössbauer Spectroscopy

The analyzed samples were homogeneous powder gently scraped from the surfaces and interfaces of the stained specimens: three tiles of Carrara and Travertine. Two samples were similarly collected from indoor specimens and from the archeological artifacts, observing the BB.CC. regulations for the sampling modality.

The absorbers were prepared by pressing finely the powder samples mixed with powder acrylic resin (Lucite) to self-supporting disks of about 10 mm diameter. Sample quantity at this stage was strictly dependent on the availability of the examined material, with the drawback of—in some cases—staying at or near the detection limit. The amount used should correspond to about 2 mg of iron oxide, with an absorption density in which the thickness does not affect the Mössbauer results. In our case, these concentrations were very difficult to reach, and to solve this problem long acquisition times (one or two weeks) were used to ensure a good signal/noise ratio in data reading, and in all cases the powders collected at the surface and interface zones were mixed and analyzed as a single sample.

Spectra were collected at 298 K (room temperature, RT) using a ^57^Co/Rh source and a conventional constant acceleration mode. A multichannel analyzer with 512 channels was used for the data recording at a range of velocity of −10 to +10 mm/s. A highly pure sample of α-iron was used to calibrate the speed, and raw data were collected in 512 channels. Spectra were elaborated by Recoil 1.04 [44] software accounting for symmetric Lorentz curves. The χ^2^ test was used to individuate the best conditions and uncertainty was obtained by the covariance matrix. A 0.02 mm/s uncertainty resulted for the isomer shift (IS), quadrupole splitting (QS), and magnetic hyperfine field (HF). Uncertainty no lower than ±3% was estimated for the doublet areas.

## 5. Conclusions

The results obtained concern the mechanisms of iron speciation and rust stain formation over the surface and inner areas of carbonate matrices of either artificially stained marbles exposed over time under different environmental conditions and archeological finds. The analytical methods adopted provided complementary information and reciprocally confirmed the overall outcome. Optical microscopy gave preliminary information and allowed the choice of the most significant samples. SEM/EDS also gave information on the diffusion mechanisms and damage of the pore net. Mössbauer spectroscopy gave information on iron oxidation states, mineral phases possibly formed as a consequence of metallic iron alteration, and insights on their aggregation state. The surface sensitivity of XPS allowed the chemical states of iron associated with those mineral phases (and possible aggregates) to be determined at a very local nanometer scale.

It was thus possible by means of combined techniques to effectively reveal the effect of oxidation, crystal growth, and hydration/dehydration reactions in the progressive transformation of iron compounds (and rust composition) from metal aggregates into mixed phases containing Fe^2+^, Fe^2.5+^ and Fe^3+^ ions, such as magnetite followed by oxide hydroxides maghemite, lepidocrocite, goethite, and as the last stage hematite.

The results obtained will have implications for the correct care of archeological artifacts, which—depending on their evolution of iron-based stains—may require different treatments for effective and safe rust removal.

## Figures and Tables

**Figure 1 molecules-28-01582-f001:**
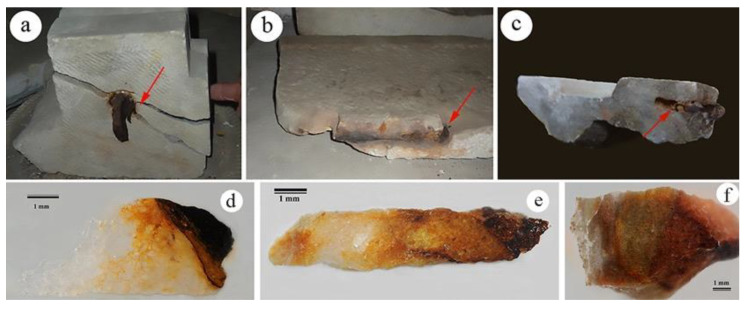
Sampling points, evidenced by red arrows, and images of cross sections acquired in reflected light (25×), respectively, for samples NS1 (**a**,**d**), NS3 (**b**,**e**) and NS8 (**c**,**f**).

**Figure 2 molecules-28-01582-f002:**
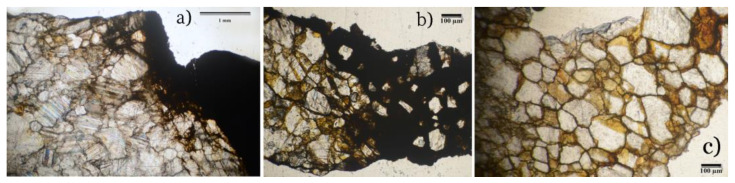
Thin NS1 (**a**), NS3 (**b**), and NS8 (**c**) sections that show the infiltration of iron phases inside the carbonate matrix. OM polarized light analysis, 25×.

**Figure 3 molecules-28-01582-f003:**
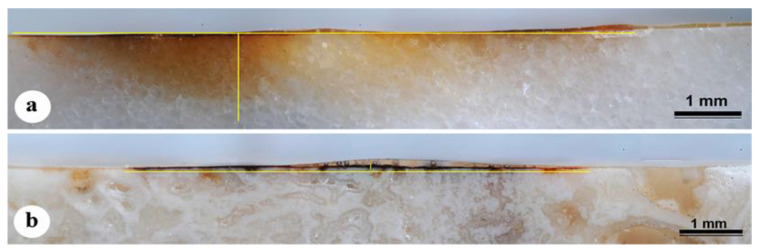
Artificially stained samples. Cross sections images acquired in reflected light (10×) of Carrara marble (**a**) and Travertine (**b**) stained outdoors for two seasons (spring and summer).

**Figure 4 molecules-28-01582-f004:**
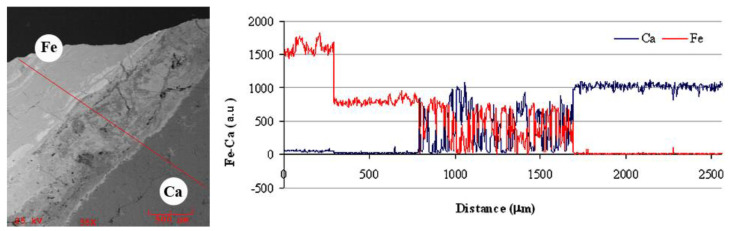
SEM image (**left**) and elemental line mapping (**right**) of iron and calcium obtained for the NS1 samples.

**Figure 5 molecules-28-01582-f005:**
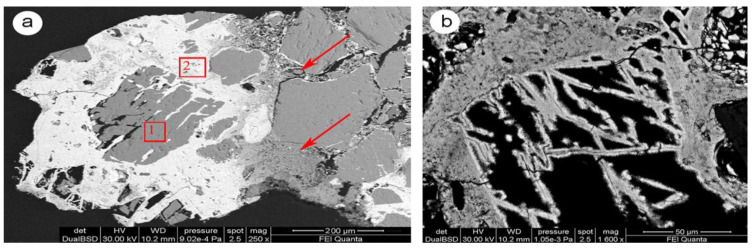
SEM image of sample NS3 showing: (**a**) the replacement of calcite (gray color) by iron rich phases (white color); (**b**) morphological presence of colloid microporous goethite and light gray crystals of lamellar hematite. Red arrows highlight the diffusion front of iron. Points 1 and 2 were analyzed by EDS (see Table 1).

**Figure 6 molecules-28-01582-f006:**
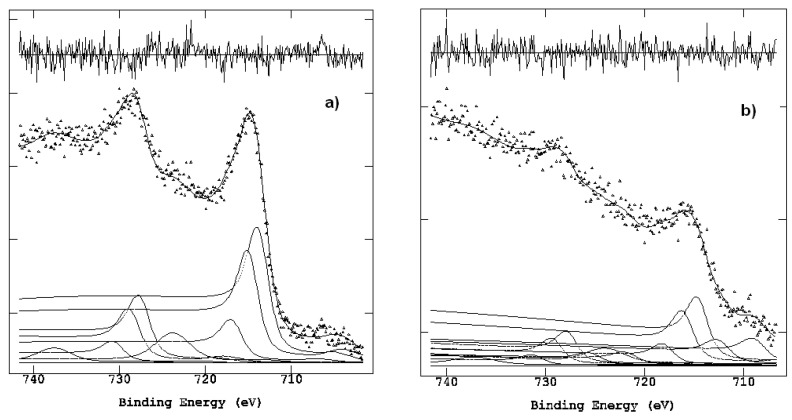
Spectra of the Fe region obtained for the stained Carrara samples with long outdoor exposure: (**a**) external growth layer; (**b**) interface zone.

**Figure 7 molecules-28-01582-f007:**
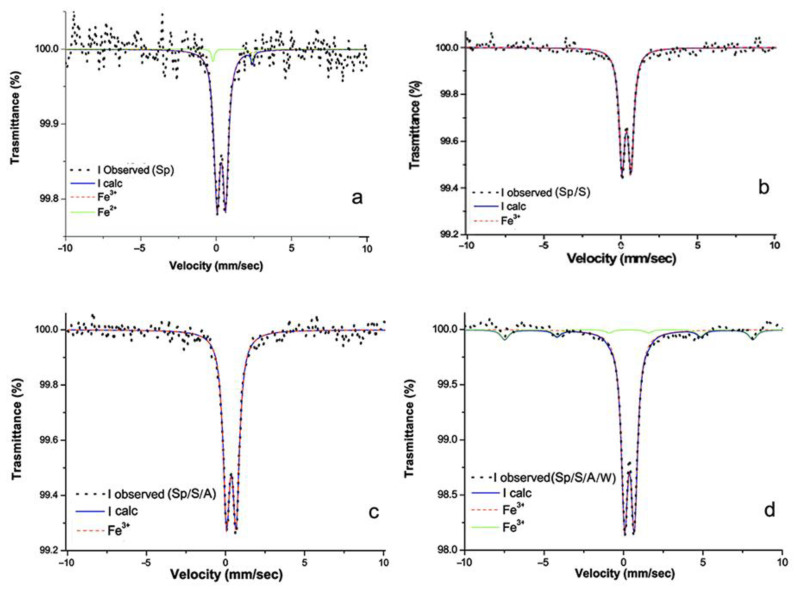
Mössbauer spectra of marble samples artificially stained outdoors during: (**a**) one season (spring, Sp); (**b**) two seasons (spring and summer, Sp/S); (**c**) three seasons (spring, summer, autumn, Sp/S/A); and (**d**) four seasons (spring, summer, autumn, winter, Sp/S/A/W). Observed absorption intensity (I obs, black dots) and calculated total intensity (I calc, continuous blue line) are plotted together with calculated absorption components (dashed red line for the Fe^3+^ central doublet, continuous green line for accessory components).

**Figure 8 molecules-28-01582-f008:**
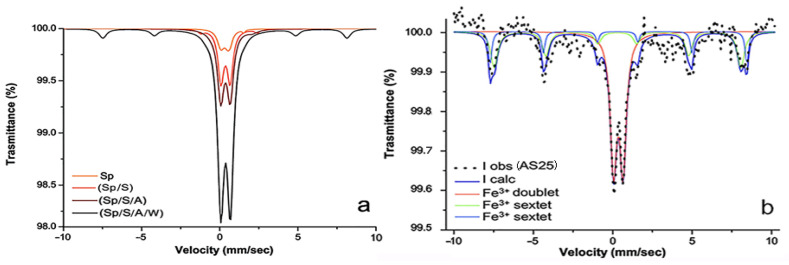
Mössbauer spectra of: (**a**) marble samples stained outdoors after exposure of one-to-four seasons (see Figure 7 for details); (**b**) sample AS25 stained for three months in a highly humid environment. Observed absorption intensity (I obs, black dots) and calculated total intensity (I calc, continuous blue line) are plotted together with calculated absorption components (continuous red line for the Fe^3+^ central doublet, green and cyan for Fe^3+^ sextets).

**Figure 9 molecules-28-01582-f009:**
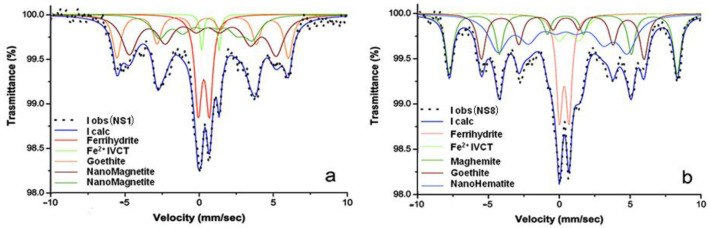
Mössbauer spectra of the historical marble samples: (**a**) NS1; (**b**) NS8. Observed absorption intensity (I obs, black dots) and calculated total intensity (I calc, continuous blue line) are plotted together with calculated absorption components (see text).

**Figure 10 molecules-28-01582-f010:**
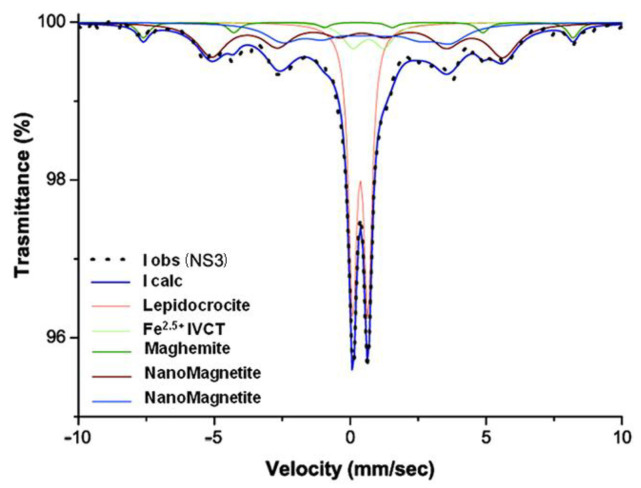
Mössbauer spectra of the historical marble NS3. Observed absorption intensity (I obs, black dots) and calculated total intensity (I calc, continuous blue line) are plotted together with calculated absorption components (see text).

**Table 1 molecules-28-01582-t001:** EDS element analysis (wt%).

Spectrum	O	Na	Mg	Al	Si	P	S	Cl	K	Ca	Fe
1	43.3	0.1	0.7	0.2	0.3	0.1	0.1	0.1	0.0	54.8	0.3
2	33.0	0.7	0.7	0.4	4.3	0.6	0.2	0.1	0.2	4.3	55.6

**Table 2 molecules-28-01582-t002:** Curve-fitting results of the XPS regions of the Carrara marble samples with long outdoor exposure. (**a**): C1s, O1s, Ca 2p (with superimposed Mg KL_1_L_2_ signal) and Mg 1s regions; (**b**): Fe2p regions.

(a)				
*Element, Orbital Chemical State*	*Carrara Surface*		*Carrara Interface*	
	Corrected BE (eV)	Normalized Area	Corrected BE (eV)	Normalized Area
C_1s_ → lower BE carbons *	283.5	122.8	283.3	247.1
C-C (IS)	285.0	1267.8	285.0	1887.4
C-O/C-O-C	286.0	946.9	286.5	328.9
O-C-O/C=O/COO	288.7	341.5	288.6	517.7
CO_3_^2−^/COOR	290.5	723.1	290.1	3075.1
* Typical of carbides, polycyclic compounds etc.				
Ca 2p_3/2_→Ca	---	---	344.8	130.1
nnaCaO/	345.9	44.8	346.7	692.3
CaCO_3_	347.9	441.7	347.9	2437.6
Ca 2p_1/2_→ Ca	---	---	348.3	65.0
CaO	349.5	22.4	350.2	346.1
CaCO_3_	351.5	220.9	351.4	1194.4
Mg KL_1_L_2_ ^*^	---	---	352.8	---
SU_1_ CaCO_3_	---	---	355.9	100.1
SU_2_ CaCO_3_	---	---	359.7	121.4
* Auger signal				
O1s-peak 1 (metal oxides)	530.4	2422.3	530.0	1275.8
O1s-peak 2 (calcite matrix)	532.3	4337.8	532.1	11610.0
Mg_1s_→ MgO	1304.8	51.8	1305.4	504.8
**(b)**				
Fe2p_3/2_ Fe |→ Metallic FeO ↔ Fe(II) compounds	---	---	704.6 708.2	30.0 28.7
Fe_2_O_3_main peak	710.7	387.1	710.2	67.0
Fe_2_O_3_ MS(I)	711.9	290.3	711.5	53.6
Fe_2_O_3_ MS(II)	713.8	130.6	713.5	21.4
Fe_2_O_3_ SU	720.6	185.6	719.6	43.5
Fe2p_1/2_ Fe |→ Metallic FeO ↔ Fe(II) compounds Fe2O_3_ main peak	--- --- 724.1	--- --- 193.5	717.6 721.2 723.5	15.0 14.4 26.1
Fe_2_O_3_ MS(I)	725.3	145.1	724.8	19.6
Fe_2_O_3_ MS(II)	727.2	65.3	726.6	9.1
Fe_2_O_3_ SU	734.2	92.8	732.6	16.0

**Table 3 molecules-28-01582-t003:** ^57^Fe Mössbauer parameters relative to the studied archeological and artificially stained marble samples.

Sample	IS (mm/s)	QS (mm/s)	HF (T)	A (%)	Attribution
**Sp**	0.34	0.57	–	100	Nanogoethite/lepidocrocite/ferrihydrite/nanohematite
**Sp/S**	0.37	0.59	–	100	Nanogoethite/lepidocrocite/ferrihydrite/nanohematite
**Sp/S/A**	0.36	0.61	–	100	Nanogoethite/lepidocrocite/ferrihydrite/nanohematite
**Sp/S/A/W**	0.37	0.61	–	89	Nanogoethite/lepidocrocite/ferrihydrite/nanohematite
	0.33	–	49	11	Maghemite/hematite
**AS25**	0.35	0.60	–	14	Nanogoethite/lepidocrocite/ferrihydrite/nanohematite
	0.30	–	48	54	Maghemite/hematite
	0.34	–	50	32	Maghemite/hematite
**NS1**	0.37	–	36	17	Goethite
	0.41	–	31	5	Nanomagnetite
	0.59.	–	20	24	Nanomagnetite
	0.31	0.75	–	21	Ferrihydrite
	0.76	1.18	–	33	Fe^2.5+^ (IVCT)
**NS3**	0.27	–	49	37	Maghemite/hematite
	0.35	–	33	5	Nanomagnetite
	0.67	–	20	6	Nanomagnetite
	0.37	0.55	–	27	Lepidocrocite
	0.67	1.15	–	25	Fe^2.5+^ (IVCT)
**NS8**	0.35	–	50	10	Maghemite/hematite
	0.35	–	36	15	Goethite
	0.35	–	28	22	Nanohematite
	0.35	0.67	–	18	Ferrihydrite
	0.67	1.44	–	35	Fe^2.5+^ (IVCT)

**Note**: Room temperature measurements. IS = isomer shift relative to α-Fe foil, QS = quadrupole splitting, HF = magnetic hyperfine field, A (%) = percentage of total absorption area. Errors are estimated at about ±0.02 mm/s for IS and QS, about ±0.7 T for HF, and no less than ±3% for both doublet and sextet areas.

**Table 4 molecules-28-01582-t004:** (**a**) Archeological samples. (**b**) Artificially stained samples.

**(a)**			
**Sample**	**Description**	**Provenience**	**Date**
**NS1**	Base of bust	Vatican Museum	19th century AD
**NS3**	Sarcophagus’ fragment	Vatican Museum	4th century AD
**NS8**	Fragment of marble slab	Roman bath Palazzo Valentini (Room 5)	4th century AD
**(b)**			
**Samples**	**Description**		
**AS25**	Travertine stained indoor for three months		
**Sp**	Marble stained outdoor during spring		
**Sp/S**	Marble stained during spring/summer		
**Sp/S/A**	Marble stained during spring/summer/autumn		
**Sp/S/A/W**	Marble stained during spring/summer/autumn/winter		

## Data Availability

Not applicable.

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
