# Peer review of "Analytical Investigation of Iron-Based Stains on Carbonate Stones: Rust Formation, Diffusion Mechanisms, and Speciation"

_molecules, 2023, doi:10.3390/molecules28041582_

Round 1
Reviewer 1 Report
The paper is very interesting and well written. It could be accepted for publication in Molecules after having considered the following comments:
Line 16 “anesthetic” Chang in “unesthetic”
I suggest to avoid to use different terms to mean the same decay pattern (such as stain and/or chromatic alteration .see Line 16); in fact chromatic alteration means something different following the main National and International accepted standards (UNI, CEN or ICOMOS Glossary). Please use stain all over the text.
Line 19. Is the term “natural” needed? Are the chelating agents “natural” in any case? The term is appropriate at Line 74.
Both reference 1 and 2, should be substituted (Line 37) with some more general on carbonate rocks decay. (The same reference could be conserved at line 72)
Reference 3 deals with copper based compounds. Please substitute with a reference on iron based compounds.
Line 144. Can the authors add some more detail about the procedure used to artificially stain the Carrara marble and the Travertine? (or refer to Line 364). May be some detail about the type of travertine is needed? Tivoli travertine?
Line 196. Change in “Carrara marble and Travertine”.
Line 360. The correct reference should be UNI 11182 (Being NorMaL the previous name). Same at Line 376
Author Response
We are grateful to the reviewer for the kind support to our work and the timely comments to be easily identified and amended. We have edited point by point as requested:
We have used only the word ‘stain’ to identify the same decay process throughout the text
We have removed ‘natural’ (related to chelating agents) at line 19 and conserved at line 74
We have removed both references 1 and 2 at line 37 (both conserved at line 72) and replaced with two more general articles on stone decays
We have not removed reference 3 but expanded the sentence to include it as another example of metals-stone interactions (from our previous work) and then added a new reference specifically related to iron, reference 4
At line 144 we have referred to Materials and Methods for the artificial staining procedure and at line 364 specified the type of Travertine (Tivoli quarries)
At line 196 we have changed the sentence in ‘Carrara marble and Travertine’
At lines 360 and 376 we have used the right reference UNI 11182
We really hope to have satisfied the reviewer requests and improved the manuscript.
Reviewer 2 Report
The current study concerns the mechanisms of the stains occurring on the carbonate stones and the identification of the stained products using various modern technologies. The identification of the stain products along the way of stain propagation is commendable but a better attempt of the delineation of stain mechanisms in view of various chemical reactions would have upgraded the quality of the paper. Have there been any effects of the purity or impurity of the pins and brackets in the reaction mechanisms? There is an impression implying the Fe/Ca ratio plays a role in the propagation of stain development. Some comments in relations to potential chemical reactions are required. Is CaCO3 soluble under the conditions such as the presence of impurities in the iron phase? For example, the presence of S would make the solution more acidic.
It would make easier for the readers to follow if Materials and Methods comes before the Experimental Results and discussion. As the order is changed, the repetitive descriptions of the samples in the Results section should be cut to avoid duplication of statements.
Line 160: It should be 1.8 mm instead of 1.3?
Author Response
We thank the reviewer for supporting our work and for addressing his first important comment on basic aspects to be considered to improve the quality of the paper. As suggested by the reviewer, we addressed some aspects of carbonate stones degradation and stain mechanisms in view of chemical reactions, in particular, on the effect of impurities in the iron phases, the influence of Fe/Ca ratio and aggressive pollutants as sulphur compounds.
To properly answer to the first comment, we have added a new paragraph ‘Discussion’ with further discussion of the obtained results following the previous Results section. In there, some related references showing that these basic aspects are amply considered in literature and taken into accounts to rationalize our results, i.e. iron chemical states evolution correlated to stains diffusion within carbonates, association of carbonaceous deposits with iron stains always detected by surface analysis. However, studies on the effect of separated variables were not considered in this work, our experiments being dedicated to the whole characterization of stained marbles and carbonate stones in view of proposing an effective procedure, suitable case by case, for their recovering (as we did in previous work for the ‘upper layers’ removal of surface-stained marbles). As added in the Introduction, the artificial staining procedure in our laboratory was meant to mirror the concomitant effects of environmental pollutants and natural weathering affecting real artifacts with time, using pure metallic iron for the staining contact (99% purity verified by XRF in the course of the triennial doctorate project)
Regarding the second requirement, at difference of the allowed addition of the paragraph ‘Discussion’, the anticipation of Materials and Methods in the manuscript is not allowed for the Special Issue editorial of Molecules. However, in the revised version, we have referred to ‘Materials and Methods’ for experimental details
Regarding the third punctual comments, at line 360 1.3 mm was replaced with 1.7 mm (after checking the original data)
We really hope to have met the reviewer requests and with new Discussion improved the quality of our paper.
Reviewer 3 Report
Paper title: Analytical investigation of iron stains on carbonate stones
In this study, the iron stains on archaeological marbles were studied and compared with those artificially produced on samples. The approach adopted was based on different techniques.
The paper could be interested for readers after addressing some major revisions:
1. Title of the manuscript should be revised.
2. Abstract is too general, please rewrite.
3. In Introduction part, please highlight the strength of the current paper.
4. Figure 1 and figure 2 could be regrouped
5. Figure 5, please mention a and b for each title
6. Page 5, avoid the points highlighting some results
7. Figure 8 and also the remaining figures are not clear. Please improve
7. Improve the style of the provided tables
8. Conclusion is too long
Author Response
We thank the reviewer for his support and for suggesting the following revisions in order to highlight the strenght of our work and improve the readers interest for our paper The changes required were all accepted except one for editorial reasons, as reported below:
The title was revised and focused on the carried work
The abstract was revised and the comparative experiments on real samples and artificially stained samples highlighted
A sentence was added in the Introduction to highlight the experimental procedure using combined techniques and the final scope of our research that is to beneft of the overall outcomes to properly plan the recovery of the cultural heritage under study. The new sentence in Introduction is complemented by the added paragraphs in Results and Discussion.
We have grouped figures 1 and 2 in only one figure, Fig.1
We have added ‘left’ and ‘right’ in the caption of Figure 5 (now Figure 4) instead of inserting a) and b) in the figure (just for limited space of insertion)
We have removed the points highlighting the results, at the end of the XPS section.
The Mossbauer figures (now from 7 onwards) and respective captions were improved and made clearer.
We have used the style indicate for this Special Issue indications for the Tables, also in this case we have decided not to proceed with changes not considered by the editorial team.
We have shortened the Conclusion and the removed paragraphs have become part of the new Discussion paragraphs
We hope to have fully responded to the reviewer requests, eventually leaving the format improvements of our long Tables at the Editorial consensus.
Round 2
Reviewer 3 Report
The paper is now improved.